# Biological Deterioration and Natural Durability of Wood in Europe

**Juan A. Martín * and Rosana López ***

Departamento de Sistemas y Recursos Naturales, ETSI Montes, Forestal y del Medio Natural, Universidad Politécnica de Madrid, 28040 Madrid, Spain
* Correspondence: juan.martin.garcia@upm.es (J.A.M.); rosana.lopez@upm.es (R.L.)

**Abstract:** In recent years, the use of wood has gained social interest, leading to a global increase in its demand. Yet, this demand is often covered by the production of woods of low natural durability against biological deterioration. The main biological agents with the potential to attack the structural integrity of wood are wood-decay fungi, saproxylic beetles, termites, and marine molluscs and crustaceans. In most circumstances, fungi are the main wood-deteriorating agents. To attack the cell wall, wood-decay fungi combine a complex enzymatic mechanism with non-enzymatic mechanisms based on low-molecular-weight compounds. In some cases, the larvae of saproxylic beetles can also digest cell wood components, causing serious deterioration to wooden structures. The impact of subterranean termites in Europe is concentrated in the Southern countries, causing important economic losses. However, alien invasive species of voracious subterranean termites are expanding their presence in Europe. Wooden elements in permanent contact with marine water can be readily deteriorated by mollusc and crustacean borers, for which current preservatives lack efficacy. The natural durability of wood is defined as the inherent resistance of wood to catastrophic action by wood-destroying organisms. Besides exposure to the climate, product design and use conditions, the natural durability of wood is key to the prediction of the service life of wooden products, which can be shortened due to the impact of global change. The major wood properties involved in natural durability are related to the composition of lignin in the cell wall, the anatomy of the xylem, nutrient availability, the amount and composition of heartwood extractives, and the presence of moisture-regulating components since wood moisture content influences the establishment of wood-degrading organisms.

**Keywords:** wood deterioration; rot fungi; decay; termites; marine borers; saproxylic insects; wood durability; wood anatomy; wood preservation; wood chemistry

## 1. Introduction

For thousands of years, humans have used wood as one of their main building materials. Nowadays, the demand for wood has decreased in some markets due to competition from other materials (e.g., PVC, fiberglass, concrete), but the overall consumption of timber is rising [1]. This trend is expected to continue growing in the future due to the relevance that the bioeconomy is gaining today [2]. In response to higher wood consumption, the area occupied by forest plantations is increasing in most developed countries, while deforestation in tropical parts of the world is still of serious concern [3]. To maximize yield, foresters often apply intensive silvicultural management to fast-growing tree species, resulting in wood with wide growth rings, lower wood density, a lower proportion of heartwood, and, in many cases, lower wood durability [4]. Although preservation or modification methods to improve the durability of wood have been developed, many make use of compounds that are toxic to the environment. While research on effective and sustainable methods is needed [5], the study of traits related to the nat-

ural durability of wood is of great importance for increasing wooden products' service life, choosing an adequate wood species for an application and increasing the service life of wooden products, in general. Furthermore, the current context of globalization and climate change is influencing the biological agents that deteriorate wood products. On the one hand, globalization in the trade of wood and wood packaging increases the probability of the accidental introduction of alien organisms, which in some cases emerge as invasive species with the potential to attack timber [6,7]. On the other hand, global warming is altering the global distribution of certain xylophagous organisms, e.g., favoring their spread to northern latitudes or higher altitudes [8,9]. Under this complex scenario, the aim of this review is to provide a general overview of the main biological agents with the potential to deteriorate wood in Europe, keeping in mind that new biological threats will possibly arise in the future. Yet, not all wood species or environmental situations have the same susceptibility to biological deterioration [10]. Therefore, we also address the main environmental and wood-inherent factors influencing the natural durability of wood.

## 2. Biological Agents Responsible for Wood Deterioration and Potential Shifts Caused by Global Change

This section reviews the main biological agents causing the deterioration of wood in service in Europe. Because some agents, particularly termites, are found within a limited geographical distribution in Europe, the risk of attack by a particular biological agent will depend on the geographical location of the wooden element [11] (European Standard EN 335:2013). The likelihood that a wood product will be attacked by a particular biological agent is also influenced by the local environmental conditions that surround the wooden element, with those factors affecting the moisture level that can be reached in wood being of particular importance (e.g., climate, soil texture, structure and organic matter content, constructive design, etc.). Furthermore, global change has significantly altered the spatial distribution and long-distance spread of forest insects and pathogens [12]. The long-distance transfer of non-native biotic agents to new locations is most notable for the emergence of various novel forest pests, which can affect wooden products (e.g., the invasive *Reticulitermes flavipes* or *Lyctus brunneus* in Europe; see Section 2.2.1). Pest and pathogen introductions have primarily been facilitated by international trade and the movement of infected live plants, wood materials, or soil [13], and the dispersal paths of biotic agents can be altered by changing climatic circumstances. Increasing temperatures affect many aspects of insect behavior, such as flight distances and the timing of seasonal emergence [14]. A number of bark beetle pests, such as *Ips typographus* and *Dendroctonus ponderosae*, have been recorded to have earlier emergences and extended flight season lengths in warming conditions. Insect and microorganism survival rates are also directly impacted by climate change, which also affects the population growth rates of the agents. Increasing winter temperatures in temperate locations have been related to changes in the prevalence and effects of a range of biotic agents because rates of overwinter survival may restrict epidemics and breakouts (e.g., [15]). Insects may be able to produce more generations in a year as a result of warming temperatures and earlier emergence periods, which could have an impact on population growth, overwinter survival, and the risk of outbreaks. Furthermore, complex mutualistic or parasitic connections with symbionts or natural enemies control the populations of many insects, and these interactions may change as climatic circumstances change [16]. It was predicted that warming and humidity would dramatically shorten the useful life of timber building materials and that the location and current climate would have a significant impact on the severity of climate-induced changes [17].

Table 1 summarizes some of the most frequent biological agents deteriorating wood in service in Europe and the estimated risk of attack according to the class of use. In the following sections, we delve into the mechanisms of wood deterioration by these biological agents. The term 'wood deterioration' is used in this review in a broad sense and

refers to any biological agent able to affect wood properties, such as mechanical or aesthetical traits. The term 'wood decay' is used specifically for the biochemical decomposition of wood by microorganisms.

**Table 1.** Frequent biological agents deteriorating wood in service in Europe.

| Wood Deterioration Agent | | Class of Use [1] | | | | |
|---|---|---|---|---|---|---|
| | | I | II | III | IV | V [2] |
| Microorganisms [3] | White and brown rot fungi | | • | ● | ⬤ | ⬤ |
| | Soft rot fungi | | | • | ⬤ | ⬤ |
| | Stain fungi and moulds | | • | ● | ⬤ | ⬤ |
| | Bacteria | | | • | ⬤ | ⬤ |
| Beetles | *Hylotrupes bajulus* (Cerambycidae) | ⬤ | ⬤ | ⬤ | ⬤ | • |
| | *Trichoferus* sp. (Cerambycidae) | ⬤ | ⬤ | ⬤ | ⬤ | • |
| | *Anobium punctatum* (Anobiidae) | ⬤ | ⬤ | ⬤ | ⬤ | ● |
| | *Xestobium rufovillosum* (Anobiidae) | • | ● | ● | ⬤ | ● |
| | *Lyctus* sp. (Bostrichidae) | ⬤ | ⬤ | ● | • | • |
| | Curculioninae (Scolytidae) | | • | ⬤ | ⬤ | ⬤ |
| Termites | *Reticulitermes* sp. (Rhinotermitidae) | • | ● | ⬤ | ⬤ | |
| | *Kalotermes* sp. (Kalotermitidae) | • | • | ● | ⬤ | |
| | *Criptotermes* (Kalotermitidae) | ⬤ | ⬤ | ● | ● | |
| Marine borers | Teredinidae | | | | | ⬤ |
| | Limnoriidae/Cheluridae | | | | | ⬤ |

Risk of attack: • = low, ● = medium, ⬤ = high

[1] The risk of attack is tentatively presented for the five classes of use as an interpretation of the authors of the existing literature in the field, their personal experience and the European Standard EN 335:2013, which defines the five classes of use. Absence of dots indicates negligible risk of attack. Black dots: risk for hardwoods and softwoods; green dots: risk mainly for hardwoods; blue dots: risk mainly for softwoods. I = wood indoors without exposure to moisture; II = wood indoors with occasional exposure to moisture; III = wood outdoors without contact with soil or water; IV = wood

in contact with soil or freshwater; V = wood submerged in saltwater. It should be acknowledged that the frequency and significance of the biological risk largely depend on the geographic location, particularly in the case of termites, which are locally present in Europe (see Figure 5). [2] The aerial part of certain elements may be subject to attack by xylophagous insects. [3] Submerged or water-saturated wood is not susceptible to attack by white- and brown-rot fungi, stain fungi and moulds but is susceptible to attack by bacteria and soft-rot fungi.

*2.1. Microorganisms*

Wood is a heterogeneous (anisotropic) material made up of different types of cells and molecules. The main components of wood are hemicelluloses, cellulose and lignin, which are mainly found within the cell wall and provide the wood with structural properties. These compounds, together with starch, pectin, simple sugars, fatty acids and proteins, which are mainly stored in parenchyma cells, are susceptible to degradation by specialized microorganisms, mainly fungi and bacteria. Different types of decay can result from combinations of wood species, microorganism species and environmental conditions [18]. Wood molecules stored in parenchyma cells, such as proteins, fatty acids, simple sugars and starch, can be metabolized by a wide range of microorganisms [19]. However, the vast amount of the carbon stored in wood is located within the cell wall in the form of a complex matrix of pectin, cellulose, hemicelluloses and lignin polymers. Many microorganisms have the metabolic potential to degrade the molecular structure of cell wall polysaccharides (cellulose, hemicelluloses and pectin). Lignin is a complex aromatic polymer [20] that preserves cell wall polysaccharides from biological decay and has other functions related to structural support and water conduction in the living plant [21]. The crystalline structure of cellulose within the cell wall [22] and the crosslinking of cell wall polysaccharides with lignin [23] make wood a recalcitrant material to decay by most microorganisms. Therefore, the few microorganisms able to degrade wood deploy highly specialized arrays of enzymes and metabolites to access resources in the cell wall [24], some of them specialized in metabolising both cellulose and lignin (or mainly lignin), while others are only able to metabolise polysaccharides. The mechanisms by which these microbes are able to attack wood are presented in the following subsections.

2.1.1. Wood-Decay Fungi

Fungi are heterotrophic eukaryotes colonizing a wide variety of terrestrial and aquatic environments, and their lifestyle can vary from parasitic (e.g., human or plant pathogens) or mutualistic (e.g., mycorrhizae in plants or mycobionts in lichens) to saprophytic (e.g., wood decomposers) [25]. Nutrition in fungi is mediated by external digestion. Fungi excrete digestive enzymes and metabolites that degrade the substrates to simpler forms and subsequently absorb the digested metabolites [26]. Under certain environmental conditions (see Section 3.3 for further details), wood-decay fungi are the most efficient wood decomposers [18], being the main deterioration agents of wood products worldwide [19]. To attack wood, decay fungi generally require high moisture contents in wood (>20%–30%), and below the fiber saturation point, water availability is insufficient for the survival of fungi [27]. Therefore, under conditions where the wood element remains dry (e.g., interior environments without moisture accumulation), the risk of attack by decay fungi is negligible. Furthermore, climate warming has the potential to drive changes in fungal community development. In fact, a number of studies have already reported higher actual or potential risks of wood decay of wooden elements due to global warming [28,29].

With the term 'wood decay', we refer to the biological processes by which cell wall components are degraded (Figure 1). The fungal decay of cell wall polymers is mediated by the action of a complex array of enzymes, most of which can be classified as either hydrolases or oxidative enzymes [22]. Hydrolases catalyse glycosidic bond cleavage in polysaccharides by a reaction with water, while oxidative enzymes are involved in the decay of lignin through oxidation-reduction reactions. Hydrolases, such as en-

do-1,4-β-glucanases, 1,4-β-glucosidase, endo-1,4-β-xylanases, 1,4-β-d-mannanases and others, catalyse the depolymerisation of cellulose and hemicelluloses [30]. Cellulose in the cell wall appears mainly in the crystalline form. However, a small percentage of non-organized cellulose chains form amorphous cellulose, which is more susceptible to enzymatic decay [31]. In the absence of enzymes capable of degrading the lignin matrix, cellulolytic enzymes are usually too large to be diffused into the cell wall through existing nanopores [19]. Therefore, certain fungi lacking ligninolytic enzymes have developed a non-enzymatic oxidation decay system involving diffusible low-molecular-weight compounds, in which a chelator-mediated Fenton (CMF) mechanism results in highly reactive hydroxyl radicals that degrade the structure of polysaccharides, and to some extent also lignin [32,33].

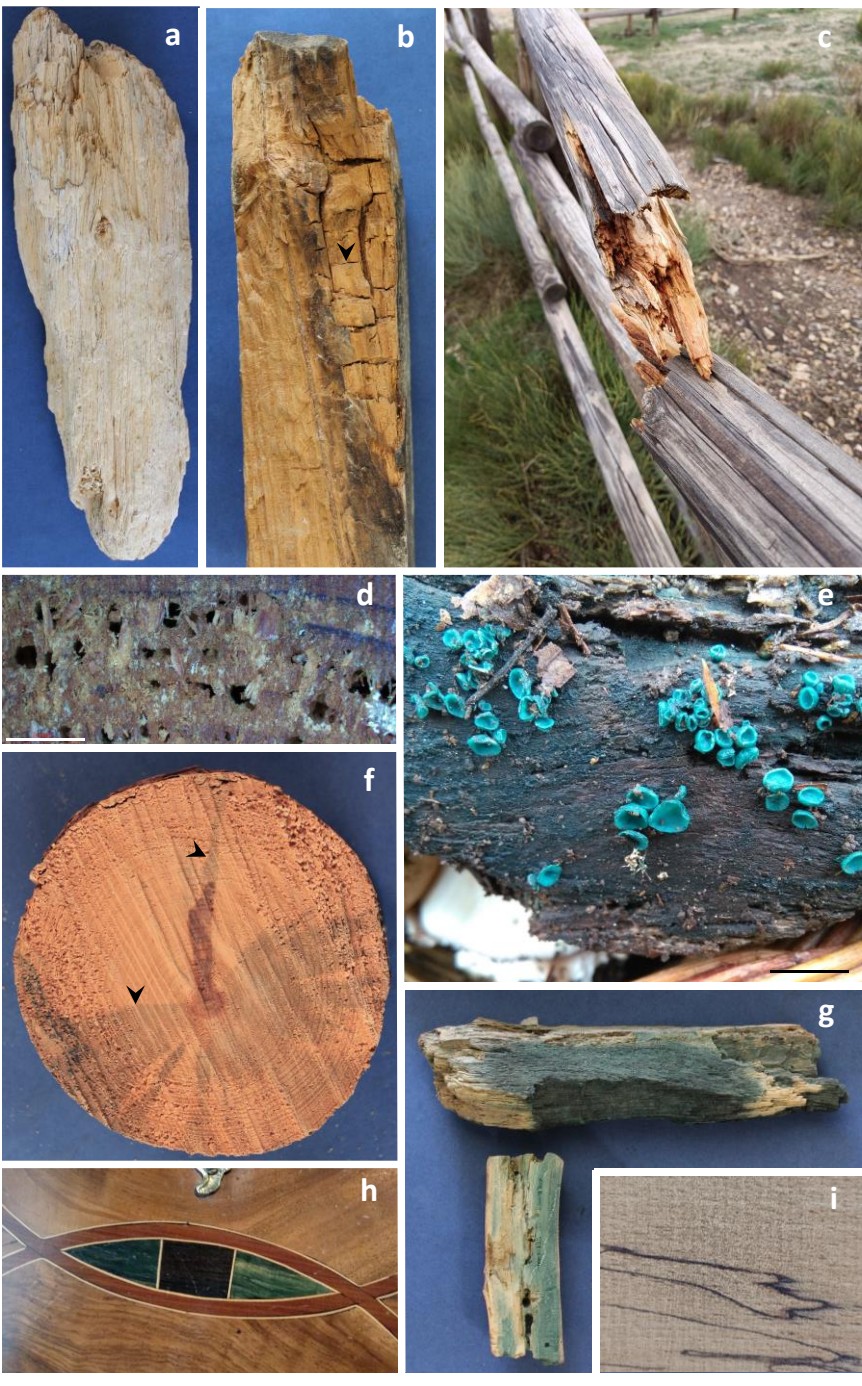

**Figure 1.** Wood deterioration caused by fungi. (**a**) Piece of wood with fibrillary appearance characteristic of white rot. (**b**) Piece of wood attacked by brown-rot fungi. Note that some cracks are formed in longitudinal and transverse planes, forming cubic-like shapes (arrowhead). (**c**) Wood railing heavily deteriorated by the action of brown-rot fungi. (**d**) Detail of the cross-section of a piece of *Pinus sylvestris* wood attacked by white pocket rot caused by *Phellinus pini*. Scale bar = 5 mm. (**e**) *Fagus sylvatica* wood showing fruiting bodies of *Chlorociboria* sp., which provides wood with a striking green colour, as can be observed in (**g**). Bar = 5 mm. (**f**) Cross-section of a pine trunk attacked by blue-stain fungi, which extend mainly in radial direction (arrowheads). (**h**) Wooden artwork in which green-stained wood by *Chlorociboria* sp. was used as decorative element (Urbino Ducal Palace, Italy). (**i**) Demarcation lines produced by fungi in wood.

Lignin-modifying oxidative enzymes include phenol oxidases (laccases) and heme-containing peroxidases (lignin, manganese and versatile peroxidases). Laccases can be considered one of the most important enzymes of the ligninolytic complex of wood-degrading microorganisms [34]. Laccases oxidize a range of aromatic compounds, including phenolic moieties of the lignin structure, aromatic amines, benzenothiols and hydroxyindols, using molecular oxygen as an electron acceptor [35]. Fungal peroxidases also catalyse the depolymerisation of lignin through oxidation-reduction reactions, during which free radicals transform the polymeric substrate into oxidized or polymerized products through hydrogen peroxide-dependent oxidation [36]. In these reactions, different low-molecular-weight, diffusible redox mediators, such as veratryl alcohol or Mn (II), can attack the phenolic units of lignin [37]. It is worth noting the important role that low-molecular-weight compounds play in all stages of wood decay, acting as diffusible oxidative agents in CMF mechanisms and electron shuttles for enzymatic systems [20].

The enzymatic and non-enzymatic wood-degrading mechanisms differ considerably among fungal lineages. Traditionally, wood-decay fungi have been classified into three main groups. The first includes the white-rot fungi, basidiomycetes, with the capability of degrading all cell wall components (polysaccharides and lignin) through an arsenal of hydrolases, oxidative enzymes, and low-molecular-weight mediators [38] (Figure 1a). Some white-rot fungi degrade all wood components simultaneously; others selectively degrade lignin but not polysaccharides, while other groups combine a selective delignification and simultaneous decay within the same substrate [39]. Furthermore, in the heartwood of living conifers, some white-rot fungi cause elongated cavities where the lignin and the hemicelluloses are preferentially degraded, resulting in the so-called white pocket rot (Figure 1d) [40]. However, this classification is not straightforward because many white-rot fungi display different types of decay [41].

The second group of wood-decay fungi includes the brown-rot fungi, basidiomycetes, able to metabolize cell wall polysaccharides but not lignin, although the arrangement of this last polymer can be modified during an attack [42] (Figure 1b,c). It seems that brown-rot fungi diverged from white-rot fungi by losing extracellular peroxidases and, consequently, the ability to degrade lignin [24]. Interestingly, evolutionary changes back to a white-rot lifestyle, from either a brown-rot or a mycorrhizal lifestyle, imply that fungi that have lost ligninolytic properties would have retained genes coding for lignin-degrading enzymes [43]. To obtain access to polysaccharides in the inner cell wall, brown-rot fungi use a CMF mechanism with the involvement of low-molecular-weight compounds, followed by enzymatic decay of the released carbohydrates. This strategy is considered to be energetically less expensive than the biosynthesis and secretion of complex enzymatic proteins [20]. While white-rot fungi affect mostly (but not only) hardwoods, brown-rot basidiomycetes are often associated with the decay of softwoods. Decay by brown-rot fungi is considered the most serious type of damage to wood in service [44].

The third group of wood-decay fungi includes the soft-rot fungi, ascomycetes, with the capacity to degrade both cell wall polysaccharides and lignin. Soft-rot fungi degrade the cell wall in the vicinity of the fungal hyphae, creating longitudinal cavities within the S2 layer of the cell wall, which provide the 'soft-wood' aspect (type 1 soft-rot fungi), or

causing the erosion of the secondary wall from the lumen and progressing toward the primary wall and middle lamella (type 2 soft-rot fungi) [45]. While the respiratory activity of white- and brown-rot fungi requires free oxygen in wood pores, soft-rot fungi are adapted to poorly airy environments. Thus, soft-rot fungi frequently attack wood in those situations where white- and brown-rot fungi cannot thrive well due to poor oxygenation, such as water-saturated wood, marine structures, railroad ties, or, in general, wood with high levels of moisture [46]. In a recent work, historic wooden structures in polar regions were found to be adversely affected by soft-rot fungi, while no wood-destroying Basidiomycota were found, suggesting the better adaptation of soft-rot ascomycetes to extreme environments [47]. Several species of soft-rot fungi have the capacity to attack wood structures, including preservative (e.g., CCA, creosote)-treated wood ([33] and refs. therein).

The traditional grouping of wood-decay fungi into these three groups is undoubtedly useful as a general way of classifying the enzymatic potential of these fungi. However, an investigation at the genome level questioned the tight classification of wood-degrading fungi into these categories, as there are fungi showing enzymatic traits belonging to different groups [48]. For example, a recent work on the enzymatic capabilities of white-rot fungi in the genus *Armillaria* revealed that members of this genus display a combination of wood decay genes that resembles the soft rot of Ascomycota [49]. This combination of genes appears widespread among basidiomycetes producing a superficial white rot-like decay [49]. Further, although the decay of wood in service by basidiomycetes can be occasionally dominated by a single species of fungus [50], the decay of wood in most situations is usually mediated by a succession of taxa with competitive, additive or synergistic interactions [51,52] and with sequential degradative activities, during which enzymatic and non-enzymatic components of the decay machinery show stage-specific expression [53]. In general, after a first phase dominated by taxa showing soft-and white-rot degrading capabilities, the decay process is preferentially driven by fungi with brown-rot degrading machinery that are stronger competitors but with weaker ligninolytic activity [54]. The black demarcation lines that some fungi exhibit in wood (Figure 1i) are used to distinguish between different species, incompatible mycelia of the same species, or mycelia that have not yet colonized wood [55]. Fungal phenol oxidases cause the lines by converting host-owned chemicals or microbial components to melanin [56].

Wood-decay fungi show varying degrees of adaptation to host species, ranging from specialists adapted to single tree species to generalists with the capacity to degrade a wide variety of substrates [57]. However, even generalist fungi, such as the white-rot *Pleurotus ostreatus*, show different decay strategies depending on host wood traits. Thus, this fungus caused cell wall erosion typical of white-rot in spruce and beech but soft-rot type I (longitudinal cavity formation) decay in oak wood [58]. Furthermore, it seems that a shift from generalists to specialists in certain fungal lineages is driven by a higher reliance on CMF mechanisms than on the enzymatic decay of wood, possibly as an adaptation to N-limited wood substrates, which hinder the performance of more nitrogen-intensive enzymatic strategy [57].

### 2.1.2. Wood Stain Fungi and Moulds

Wood stain fungi and moulds are a polyphyletic group taxonomically placed in the Ascomycetes or that have yet to find a place in the modern fungal classification as they are imperfect fungi (i.e., fungi in which the teleomorph is unknown to science). Stain fungi colonize wood mainly through ray parenchyma cells in the sapwood, feeding on easily assimilable substances such as sugars, lipids or proteins. They are often vectored by wood-boring beetles, which breed in the wood or bark of weakened, dying, or recently felled trees [59], although some of these beetle species can also attack healthy living trees [60]. Stain fungi often hasten the death of trees attacked by insects [61] and proliferate in recently felled wood or wood with high moisture contents (between 30% and 130%) [26].

Wood stain fungi do not degrade or cause only minimal cell wall attacks, and therefore, most of the mechanical properties of wood are not significantly affected by them [62]. However, they can seriously depreciate the value of wood due to colour changes, with greyish blue being the typical coloration of wood attacked by stain fungi in the genera *Ophiostoma, Grosmannia, Ceratocystiopsis, Ceratocystis*, among others (Figure 1f), although other colorations are also frequent. Blue-stain fungi discolour timber as a result of microorganism-own pigments such as melanin [63]. Fungal species in the genus *Chlorociboria* (Ascomycota) discolour branches and other woody debris in the forest (mainly in hardwoods) frequently with previous decay by white-rot fungi [64], providing wood with a striking green colour (Figure 1e,g). The wood decay by white-rot fungi is neither enhanced nor inhibited by the presence of *Chlorociboria* [64], while the green discoloration is due to a pigment (xylindein) released by the fungus [65] and deposited in the ray parenchyma cells and in vessels and fibers adjacent to the rays. This discolored wood has been historically used in marquetry and veneering [66] (Figure 1h).

Moulds are fungi that develop on damp surfaces of diverse materials, often in environments with high air humidity (around 95%), warmth and insufficient ventilation [67]. In wood, moulds usually colonize fresh cuts after tree felling, particularly on the moist sapwood, lumber or wood products (e.g., piled wood chips) stored in places with low ventilation, airtight sealed wood or wood products (such as plastic-coated paneling). They scarcely penetrate into the inner parts of the wood, and, as stain fungi, do not represent a threat to the mechanical strength of the wood [68]. However, they can diminish its aesthetic value and hamper some technological processes of wood (e.g., drying). They usually grow quickly on the wood surface and color the wood by developing pigmented spores (usually black or green) on the surface. Some of them can be harmful to humans due to their capacity to produce mycotoxins [69].

### 2.1.3. Bacteria

Bacteria are single-celled prokaryotic organisms found in almost all parts of the Earth. Some specialized bacteria are able to degrade wood but at a much lower rate than saproxylic fungi. Under most circumstances, their importance as wood-degrading agents is lower than fungi, but in certain environments (e.g., under hypoxia or anoxia), they can represent serious threats to wooden structures [70]. Many bacteria living on wood consume easily accessible compounds such as monosaccharides and pectin and may facilitate the subsequent activity of wood-decay fungi [71] but do not seriously affect the mechanical strength of wood [72]. Some species of bacteria are able to degrade wood polysaccharides in non-lignified tissues, such as the pit membranes, when wood is soaked in water [73]. Interestingly, the bacterial decay of pits can be intentionally used to improve the impregnability of wood, particularly of refractory species such as spruce [74].

A few specialised bacteria can also degrade lignified wood cell walls and can affect the mechanical properties of wood. These can be subdivided into erosion bacteria, tunneling bacteria and cavity-forming bacteria, according to the micro-structure of their attack [75,76]. Erosion bacteria attack the S3 layer of the cell wall from the lumen and then progressively enter the S2 layer, but even in advanced decay stages, the middle lamella remains intact. They can act under anaerobic conditions when other organisms, such as white- or brown-decay fungi, cannot survive. For instance, erosion bacteria were identified as the only organisms causing wood decay in wooden foundation piles from Gothenburg's historic city center, piles that were inserted into glacial clay and below the groundwater level [77]. Tunneling bacteria attack the S2 layer, creating cavities occupied by a single bacterium which, by division, increases the number of tunnels in a branching system. Tunneling bacteria were found, together with soft-rot fungi, degrading wood submerged in the coastal waters of the Antarctic, where the more efficient marine xylophages, crustaceans and molluscs, are absent [78]. Cavity bacteria also attack the S2

layer, forming diamond-shaped cavities at right angles to the axial direction of the wood, possibly involving the release of diffusible enzymes [79].

Bacterial variety seems to increase as the wood decomposes [80]. However, the knowledge of saproxylic bacteria ecology and dynamics during wood decomposition is rather limited [81]. Current evidence suggests that the presence and identity of wood-decay fungi strongly influence the bacterial community in wood [82], but the form in which these interactions develop is still largely unknown. Furthermore, recent research highlighted the complementary roles played by fungi and bacteria in wood decomposition. While fungi were found to dominate the decomposition of the recalcitrant fractions of deadwood, several bacteria participate in N accumulation in deadwood through N fixation, improving the nutritional status of deadwood for fungi [83].

### 2.2. Insects

This section is focused on the main taxa of insects able to deteriorate and devaluate wood used for construction, poles, floors, facades, furniture or artworks, among others. The insects that attack the wood of living trees are out of the scope of this work. However, we will pay attention to some insects attacking weakened or recently dead trees that transmit stain fungi because they can strongly devalue subsequent wood products and are therefore of concern for the wood market. Two groups of insects stand out for their economic impact in Europe: beetles (Coleoptera) in the families Cerambycidae, Ptinidae, Bostrichidae, and Curculionidae, and termites (Blattodea) in the families Kalotermitidae and Rhinotermitidae [26]. Other insects causing lower economic impacts are found in the orders Hymenoptera (Siricidae, Formicidae, Apidae) and Lepidoptera (Cossidae). They also can bore large galleries in wood [84] (Figure 2). However, they normally attack weakened or recently felled trees and more rarely attack wood in service [85].

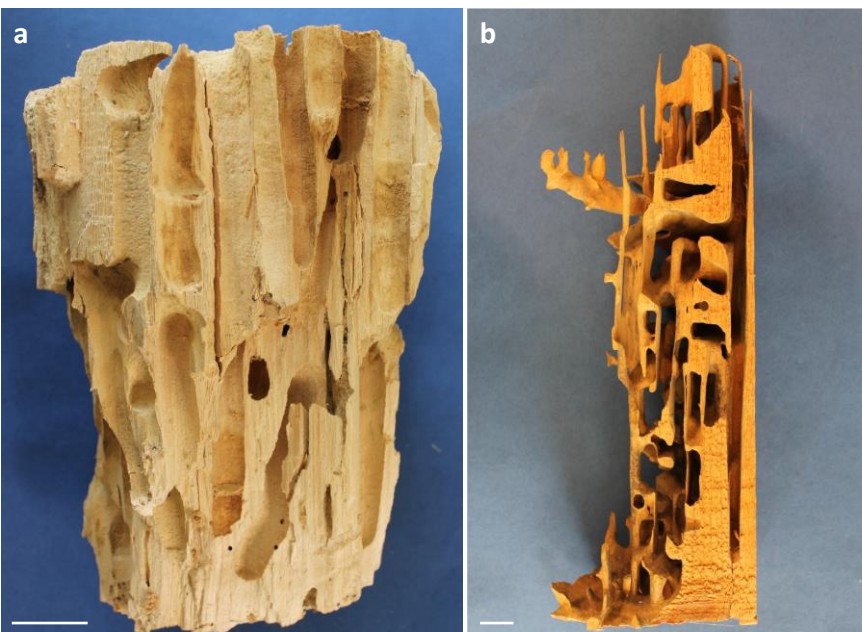

**Figure 2.** Damages caused by members of Hymenoptera in wood. (**a**) Breeding galleries bored by *Xylocopa* sp. (Apidae) in poplar wood. Bar = 2 cm. (**b**) Nest built by *Camponotus* sp. (Formicidae) in a wooden element. Bar = 2 cm.

The risk of a certain wood structure being attacked by wood-damaging insects depends on different factors, such as the intrinsic composition of the wood (e.g., starch or extractive contents) and the influence of environmental factors, mainly moisture and temperature [86]. Moreover, synergisms and antagonisms between different insect species and between insects and microorganisms can influence the susceptibility of a wood

product to insect attack [26]. The following sub-sections present the capacity of beetles and termites to attack and degrade wood components.

2.2.1. Wood Borer Beetles

The larvae of certain saproxylic beetles feed on the components of wood (cellulose, hemicelluloses, pectin, starch, simple sugars, proteins, and other trace elements), creating galleries in wood that can compromise its mechanical strength or aesthetic value [11,50] (Figure 3). However, some mycophagous species of ambrosia beetles (Scolytidae, see below) feed exclusively on their symbiotic fungi that coat the breeding galleries [87]. Beetles can damage solid wood and plywood but not fibreboard, particleboard or chipboard (European standard EN 351-3:1995). Insects in the suborder Polyphaga, which includes most wood-damaging beetles, have an arsenal of endogenous plant-cell-wall-degrading enzymes, some of which have been acquired from fungi or bacteria by means of horizontal gene transfer [88]. Although the contribution of gut microorganisms for the complete digestion of plant cell walls in these insects cannot be discarded [89,90], it seems that their digestion of cell wall polysaccharides mainly relies on endogenous genes [91]. The midgut microbiome can, however, provide nutrients that beetles cannot obtain from wood [92]. Adult saproxylic beetles usually lay their eggs in protected areas in wood, such as small cracks and slots, or bore breeding galleries for laying. Depending on the insect species, the wood's nutritional value and the environmental conditions, the larval stage can last from a few months to several years. Once the larval stage is completed, the insect pupates to give rise to the adult insect, whose lifespan is usually much shorter (several weeks of life) than the larval stage [50].

The largest wood-damaging beetles in Europe belong to the family Cerambycidae, known as long-horned beetles, with adult body sizes of around 1–5 cm and large, curved antennae [50]. In most situations, long-horned beetles only attack the sapwood, creating wide oval galleries whose section can exceed 1 × 0.5 cm (Figure 3a) and are filled with dust excrements of cylindrical shape (Figure 3b). The species with the greatest impact on processed wood in Europe is *Hylotrupes bajulus*, known as the old house long-horn borer. It usually attacks the sapwood of the coniferous genera *Pinus*, *Picea* and *Abies*, mainly in roof spaces, where summer temperatures reach levels that allow for egg laying and flight, and attacks predominately happen in the first 80 years of timber use [93]. This species can attack fairly dry wood species (a wood moisture content as low as 10%–12% is sufficient for larval growth; [94] during the 3–5 years of larval development in wood. A recent study evidenced that *H. bajulus* larvae degrade cellulose and hemicelluloses but not lignin [95]. It has been shown that the monoterpene content of wood influences the resistance of different pine provenances to *H. bajulus*, as provenances with higher monoterpene content tend to show a lower number of eggs laid by the female beetle [96]. The impact of the house longhorn beetle in Europe has, however, declined in the last decades, possibly due to the wider use of preventive treatments, the change in building practice (e.g., better isolations under roof spaces) and the decreased risk of introducing larvae through already-infested timber [94]. Other Cerambycidae species affecting European woods are members of the genus *Trichoferus* [97], attacking fences, poles, roofs and furniture of dry hardwoods, and *Ergates faber*, present in the central and eastern parts of Europe, the Caucasus and the Mediterranean (Northern Africa, Asia Minor), attacking wooden utility poles (telecommunication, lighting), railway sleepers and other wooden structures of conifer wood [98]. The impact of global change on long-horned beetles' distribution is uncertain, but Zhou et al. [99] predicted the potential global distribution of the Asian long-horned beetle *Anoplophora glabripennis*, which is an important wood-boring pest that has caused substantial damage to broadleaf trees in Asia, North America, and Europe. They concluded that climate suitability would rise in areas north of 30° N and fall in the majority of areas south of 30° N as a result of climate change.

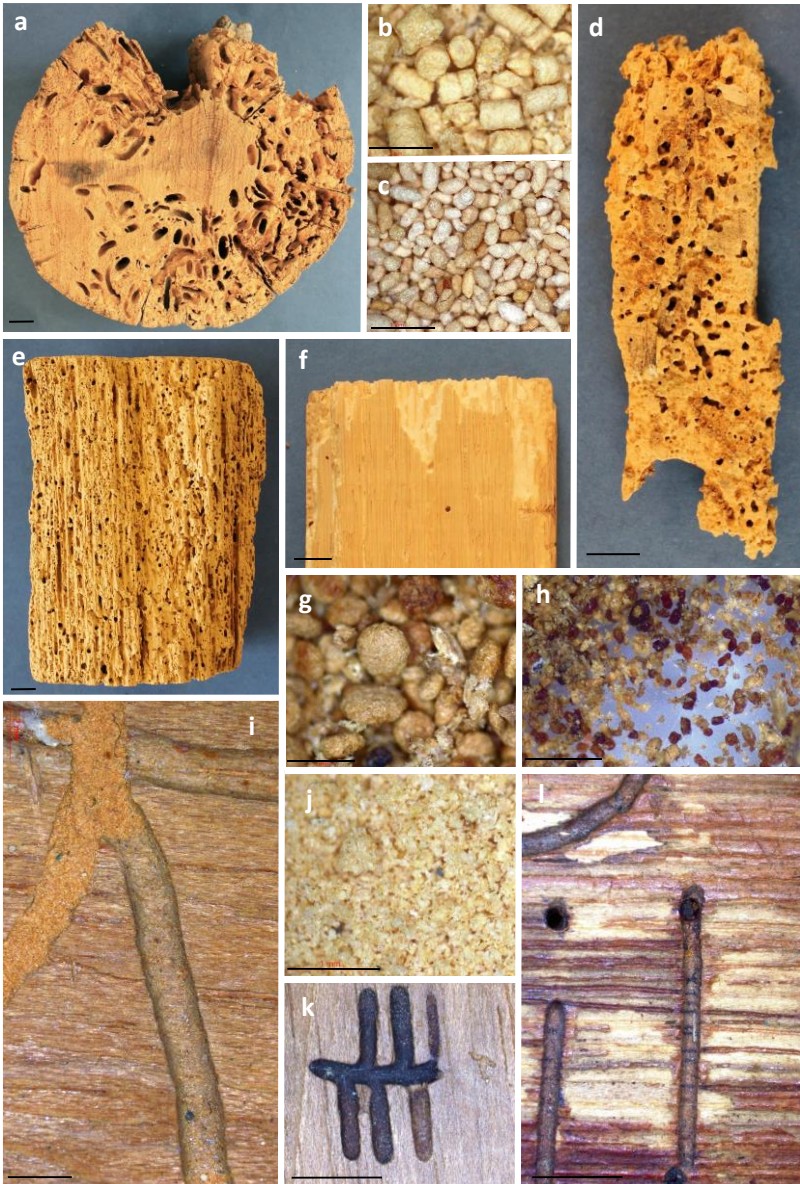

**Figure 3.** Galleries, exit holes, and dust excrements of several widespread wood-deteriorating beetles in Europe. (**a**) Cross-section of a utility pole with oval galleries bored by *Hylotrupes bajulus* (Cerambycidae). Note that galleries are present in sapwood only. Bar = 10 mm. (**b**) Cylindrical dust excrements characteristic of Cerambycidae. Bar = 1 mm. (**c**) Dust excrements of *Anobium punctatum* (Anobiidae). Bar = 1 mm. (**d**) Piece of a wooden pillar attacked by members of Curculioninae (Curculionidae). Bar = 10 mm. (**e**) Piece of wood heavily attacked by members of Anobiidae. Bar = 10 mm. (**f**) Piece of a skirting board attacked by members of Lyctinae (Bostrichidae). Note the presence of longitudinal galleries covered by dust. Bar = 10 mm. (**g**) Dust excrements of *Xestobium rufovillosum* (Anobiidae). Bar = 1 mm. (**h**) Dust excrements characteristic of Curculioninae (Curculionidae). Bar = 500 μm. (**i**) Detail of a larval gallery bored by members of Lyctinae (Bostrichidae) and partially filled with dust excrements, which are soft to the touch. Bar = 2 mm. (**j**) Detail of the amorphous dust excrements characteristic of Lyctinae (Bostrichidae). Bar = 1 mm. (**k**,**l**) Galleries and exit holes bored by members of Scolytinae (Curculionidae). Note the dark staining of the galleries as a consequence of the growth of ambrosia fungi. Bar = 10 mm.

The family Ptinidae includes in the subfamily Anobiinae the woodworm beetles, insects whose larvae can reach 6–10 mm in length before pupation, while the adult insects are small, circa 2–8 mm in length [100]. Woodworm beetles damage wood by creating circular galleries of 1–4 mm in diameter (Figure 3e). *Anobium punctatum* is the most frequent woodworm beetle attacking processed wood in Europe. It is frequently found in

furniture, doors, windows, floors, beams on ceilings, etc. Larvae attack hardwoods and softwoods with low moisture content (more than 10% [101]) and have a generation time span of 1–3 years. This beetle digests wood polysaccharides by secreting a complex battery of enzymes with cellulase, xylanase and amylase activities [102]. Furthermore, their digestive system contains symbiotic yeasts that help to digest cellulose [50]. *A. punctatum* galleries are filled with bore dust excrements, which usually have the form of an egg or a lens (Figure 3c). It has been shown that even at important levels of pine wood decay by *A. punctatum*, the loss of the wood's mechanical properties is small to moderate [103]. *Xestobium rufovillosum* (known as 'death watch beetle') is another member of Anobiidae of importance in Europe. It attacks hardwoods, preferably with high moisture contents or previously attacked by wood-decaying fungi. In fact, it is attracted to wood by volatiles generated by wood-decaying fungi [104]. *X. rufovillosum* prefers ovipositing on old wood dating from the 13th to 19th centuries rather than new wood from the 20th century [105]. The dust excrements of this beetle have a characteristic lentil-like shape (Figure 3g).

The family Bostrichidae, subfamily Lyctinae, includes the so-called 'powder-post beetles'. Adults reach a size of 2–7 mm in length, and they attack hardwoods with low moisture content (8%–20%), wide vessels, and high starch content (e.g., the sapwood of oak and tropical species), which forms the basis of their diet [100]. In fact, these insects taste wood and selectively lay their eggs in starchy wood [106]. Oviposition takes place in the vessels or pores of the wood through the introduction of a long and flexible ovipositor into the pore cavity [107]. They are frequently found attacking wood parquets, baseboards, door and window frames, creating cavities of circular sections (1–1.5 mm in diameter) (Figure 3f,i) filled with a fine dust of amorphous shape (Figure 3j) and soft to the touch. Two members of this subfamily are frequently found in Europe: *Lyctus linearis*, native to central Europe, and *L. brunneus,* an invasive species probably native to a tropical region [108]. These species have widely expanded their distribution through the international and domestic human transportation of infested wood and wood products [7].

The family Curculionidae includes the so-called 'weevils', the most diversified group of insects with more than 62,000 species known to science [109]. Three subfamilies are important for the market of wood products, although they usually do not attack sound, dry wood. Some members in the subfamily Curculioninae attack processed hardwoods and softwoods with wood moisture contents above 20%–30%. However, they require a previous attack and the partial decay of wood by rot fungi, which suggests a symbiotic relationship between them [110]. The larvae are 2–7 mm in length, and their galleries resemble the galleries bored by members of Anobiidae in size and shape (Figure 3d). Their galleries are filled with dust excrements similar to excrements of *A. punctatum* but with a more heterogeneous shape (Figure 3h). Adults can be frequently observed within degraded wood, as they are able to breed and complete their biological cycle without leaving the wood. Of special interest in the deterioration of marine timbers is the weevil *Pselactus spadix*, which can be found attacking wet wood in coastal areas, even in contact with or submerged in marine water [111,112]. Other weevils of importance for processed wood are the ambrosia beetles in the subfamilies Scolytinae and Platypodinae. These beetles bore breeding galleries in the sapwood or heartwood of dying or recently dead trees (green wood) to lay their eggs [113], and cultivate fungal spores, often of the order Ophiostomatales (Ascomycota) [114], that are carried either in the gut or in specialised spor—carrying organs (mycetangia or mycangia) [115]. Brood galleries are then colonized with ambrosia fungi, which serve as food for the offspring. Ambrosia beetles do not digest wood components but live in a nutritional symbiosis with the fungi they carry. In general, ambrosia beetles are not able to attack processed, dry wood. However, their galleries, together with the dark staining of wood caused by the ambrosia fungi (Figure 3k,l), make wood unsuitable for certain uses, depreciating its value.

### 2.2.2. Termites

Termites are a monophyletic group of eusocial insects in the epifamily Termitoidae, order Blattodea. They show heterometabolism, in which the young nymph is morphologically similar to the adult, and no pupal stage precedes maturity as they undergo incomplete metamorphosis [116]. Termites live mainly in tropical and subtropical regions, but some species live in the warmer regions of Europe and North America. Termites feed mainly on lignocellulosic substrates of plants, decisively contributing to nutrient cycling and soil formation in the forest [117], among other ecosystem services they provide [118]. There are termites that feed on litter in soil, grass or wood. Around 80 species of termites, out of the more than 2000 species described, can be considered harmful to processed wood [26,118] (Figure 4).

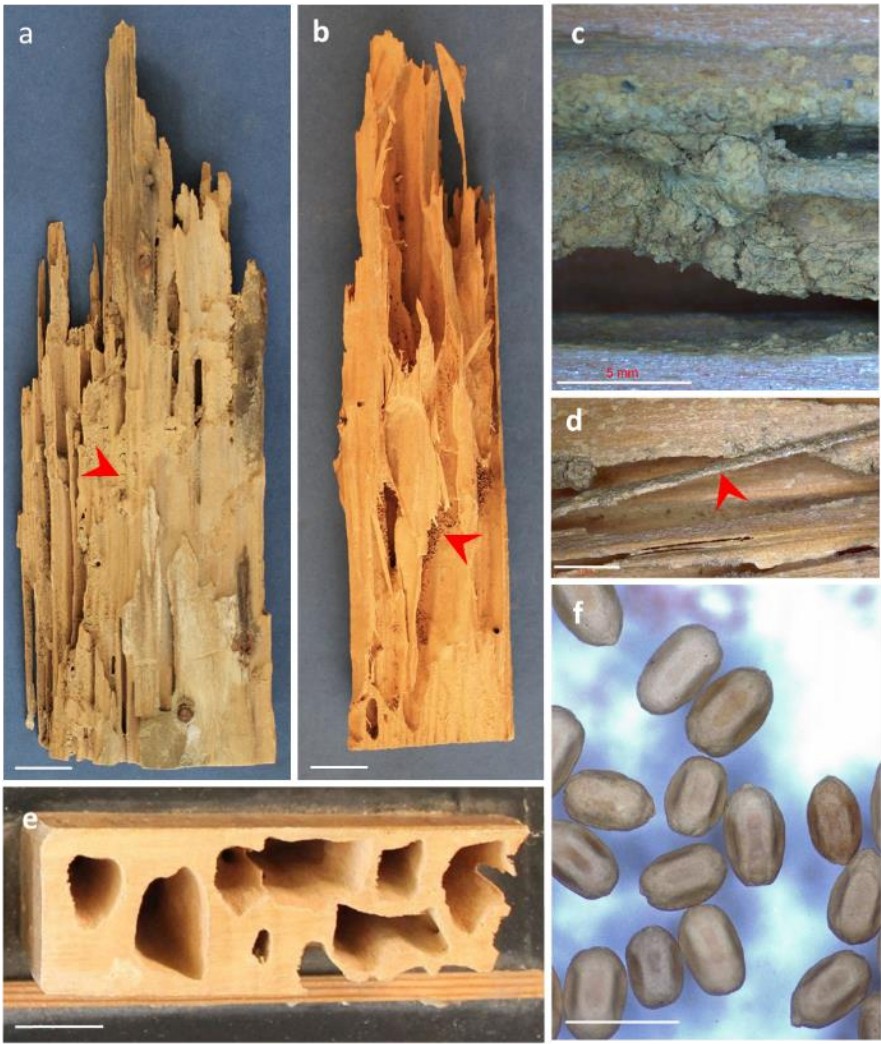

**Figure 4.** Features of wood attacked by termites present in Europe. (**a**) Wooden piece attacked by subterranean termites (Rhinotermitidae). Note the presence of mud in the galleries (arrowhead). Bar = 2 cm. (**b**) Wooden piece attacked by dry-wood termites (Kalotermitidae). Note the presence of dust excrement in the galleries (arrowhead). Bar = 2 cm. (**c**) Detail of a gallery of subterranean termites covered by mud. Bar = 5 mm. (**d**) Thin layer of wood (arrowhead) usually left by subterranean termites during their attack. This layer usually corresponds to latewood. Bar = 5 mm. (**e**) Cross-section of a wooden piece attacked by dry-wood termites and showing internal galleries. Bar = 2 cm. (**f**) Dust excrements of Kalotermitidae in the form of hexagonal prisms. Bar = 1 mm.

The social arrangement of termite colonies is divided into different castes. All members of the colony, including undifferentiated immatures, cooperate in a highly in-

tegrated manner [119]. Some individuals are infertile workers and soldiers, while others are fertile, reproductive individuals. Workers are the most numerous members of the colony and have the role of finding food sources and feeding all members of the colony by means of an oral and proctodeal (anal to mouth) nutrient exchange between individuals (trophallaxis) [50]. The proctodeal trophallaxis seems to be an important mechanism to transmit symbiont microorganisms among the colony members [120], symbionts that help the digestion of wood components (see below). In addition, workers take care of the brood and build the nest. Soldiers have the role of protecting the colony against enemies. They possess large heads with specialized defensive organs, such as large mandibles or glands that secrete defensive substances [50]. The role of the reproductive individuals is to procreate all the members of the colony. The new winged reproductives formed in the colony fly outside the nest to reproduce after synchronous mating flights and found new colonies. Flights occur in response to specific climatic conditions depending on the species [121]. Supplementary reproductives (or neotenics), either brachypterous (with rudimentary wings) or apterous, have the role of supporting primary reproductives in the procreation activity [119] and originate from nymphs or workers, particularly after the death of the primary reproductives, or in those cases in which a group of individuals moves away from the nest and loses influence from the original progenitors.

Termites degrade wood components by means of the production of endogenous cellulolytic enzymes, but symbiotic microorganisms (bacteria, protists, fungi and archaea) present in their digestive system are important for the complete digestion of wood components [122] to the extent that termites cannot live without them [123,124]. Microbial symbionts contribute to overcoming wood recalcitrance and low nitrogen availability, not only through mechanisms to digest wood components but also through their biosynthetic capacities to provide nutritional resources [125]. Termites can attack hardwoods and softwoods and usually prefer feeding on earlywood, leaving intact thin layers of latewood (Figure 4d) due to its higher density (small lumens and thick cell walls). They can destroy not only solid wood but any other product derived from wood, such as fibre- and particleboard or paper. They can also destroy plastics and other construction materials. Some of the most evolved groups of termites (Termitidae) in the tropics cultivate fungi in their nests to degrade lignocellulosic substrates [126].

Subterranean termites (Rhinotermitidae) are the most widely distributed family of termites in Europe [116]. Several native and invasive species of subterranean termites of the genus *Reticulitermes* have been identified in Europe, being the most abundant termite genus in the continent [127]. The distribution of the main native *Reticulitermes* species in Europe, and the foreseen shift in the population of the introduced *Reticulitermes flavipies* due to climate change, are shown in Figure 5. Subterranean termites form large colonies (from several hundred thousand to millions of individuals) and can destroy dry wood structures (Figure 4a). They build their nests in the soil at 70–100 cm deep and explore wood sources in their surroundings, usually within a radius of less than 10 m, with the exception of introduced populations of *R. flavipes,* with foraging ranges of up to 100 m or more [119]. Because they require high levels of moisture, damage by subterranean termites is usually associated with high moisture contents in buildings and can be found frequently in basements or in other floors as a consequence of failure in sanitation and canalization systems. For the same reason, workers usually cover the galleries in wood with moist mud (Figure 4c). Subterranean termites can be frequently found in areas and cities where they are not naturally present due to extensive human-mediated dispersal [128]. Subterranean termites frequently attack wooden structures, with a high economic impact, estimated to be approximately $11 billion per year in damage and control costs in the United States alone [129] and around $40 billion in the world [130]. With a lower economic impact, there are also several non-subterranean species of dry-wood termites in Europe (Kalotermitidae), some of which are alien invasive species (e.g., *Cryptotermes brevis*) [131]. Dry-wood termites build their nests in wooden elements, forming small colonies (hundreds to thousands of individuals) (Figure 4b,e), leaving in the cavities a

characteristic dust frass in the form of hexagonal prisms [132] (Figure 4f). They can live within structural timber and furniture inside buildings with very low moisture content and develop entirely within the wood [133]. Some species of Kalotermitidae, such as *Kalotermes flavicolis* are serious pests of woody crops and living trees [132].

Buczkowski and Bertelsmeier [134] modeled the future distribution of 13 of the most serious invasive termite species and indicated that, regardless of the climate condition and year, it is anticipated that the global distribution of most termite species will greatly rise. Instead of a straightforward poleward expansion, the range shifts by species showed a complicated pattern of distributional changes across latitudes. The tropics are possible invasion hotspots, but Europe, which was overall not among the most important invasion hotspots, showed a great increase in the number of potential invaders. There is proof that the warming environments brought on by climate change not only alter the range of invasive termite species but also encourage hybridization between invasive termite species, producing hybrid colonies with twice the growth rate of incipient conspecific colonies [135].

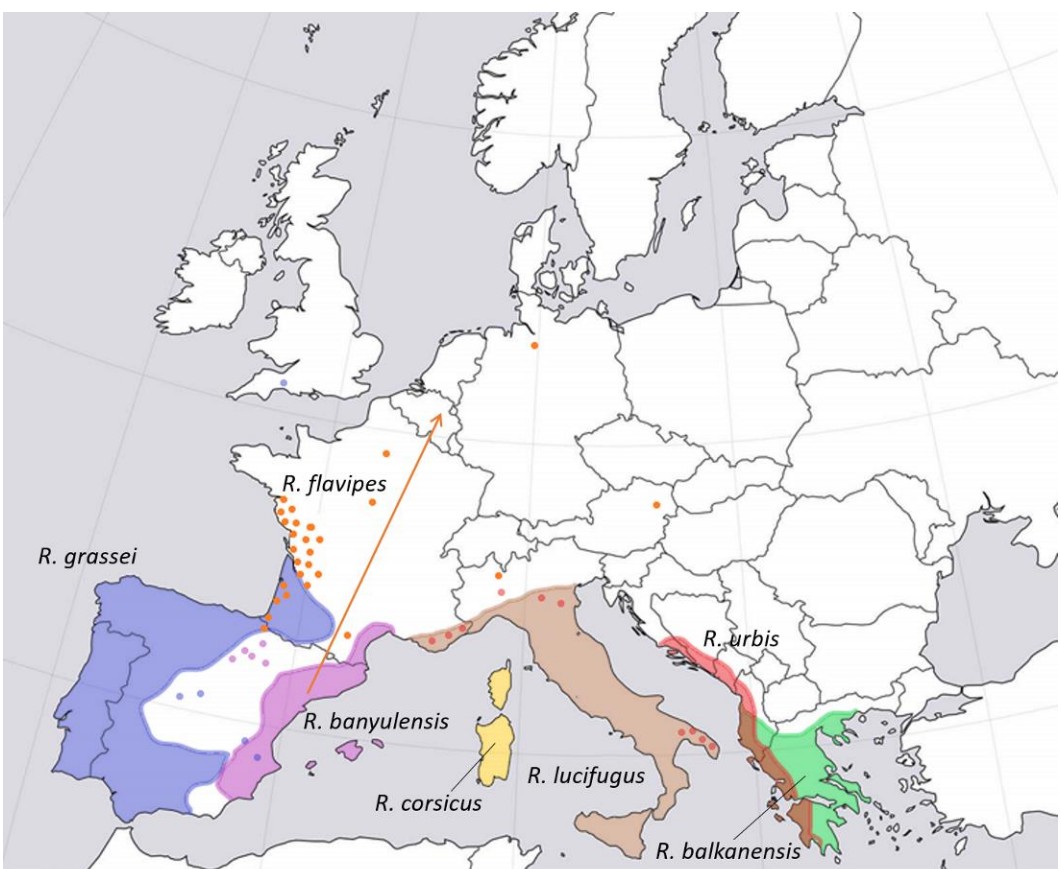

**Figure 5.** Approximate distribution of the main species of subterranean termites of the genus *Reticuliermes* present in Europe. All species are considered native except for *R. flavipes*, which is native to North America. Dots represent introductions, and colored areas represent natural distribution ranges. The orange arrow represents the shift vector of the center of gravity of the potential distribution of *R. flavipes* due to global change. Adapted from [127,134,136].

### 2.3. Marine Wood Borers

Although certain bacteria and fungi can degrade wood in marine environments, marine wood borers (molluscs and crustaceans) are the organisms that degrade submerged wood at higher rates [137] (Figure 6). They can deteriorate wharves, groynes, lock gates, house stilts, boats, floated timber, aquaculture facilities and other structures [138], being particularly destructive in warm, tropical waters. Marine wood borers can be

classified into two groups; molluscs in the class Bivalvia (families Teredinidae, Xylophagaidae and Pholadidae), known as 'shipworms', and crustaceans in the order Isopoda (Limnoriidae and Sphaeromatidae) and Amphipoda (Cheluridae), known as 'gribbles'. While shipworms can live in waters with salinities exceeding 5–9 parts per 1000, gribbles can only live in salinities exceeding 16–20 parts per 1000 [139]. In most European maritime sites, teredinids represent the most significant hazard, higher than limnoriids. However, in the Terceira Azores and Olhão (Portugal), the severity of attacks of *Limnoria tripunctata* was higher than that of teredinids [140]. Several species of marine borers present in European coasts are warm-water species. Therefore, their activity might increase in the future due to global warming [140].

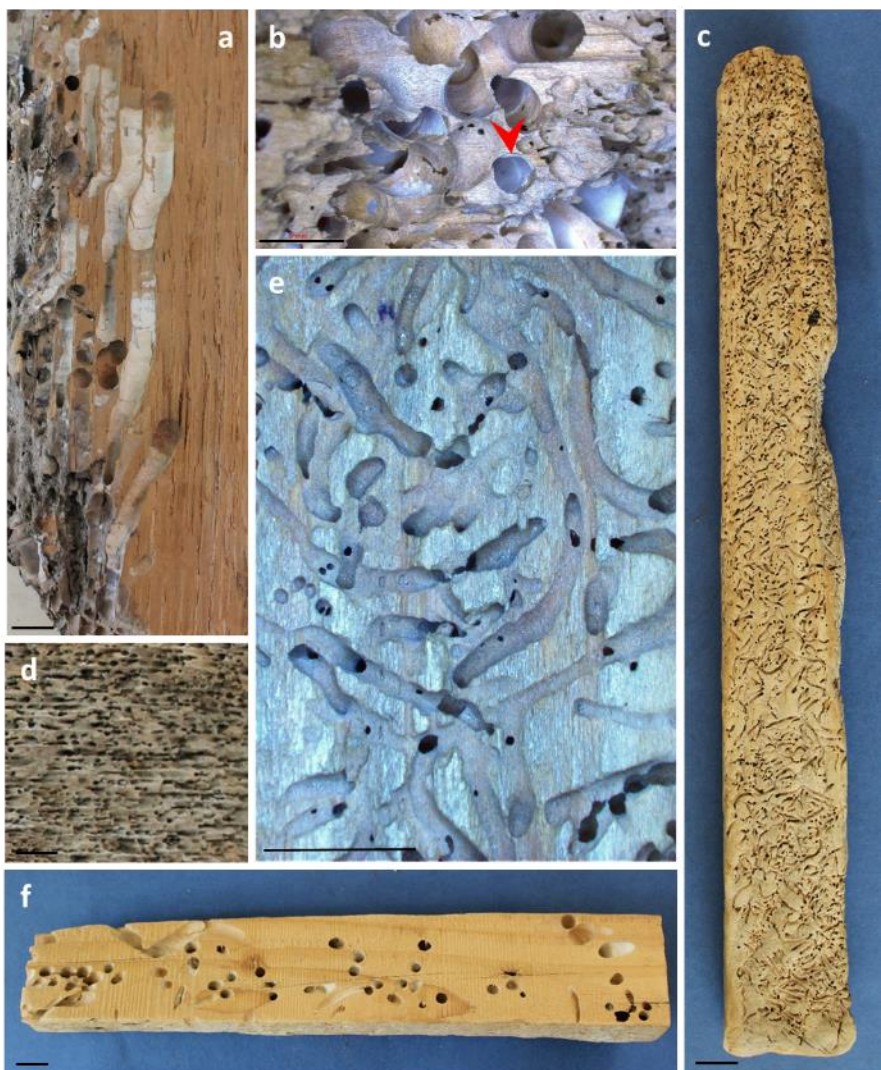

**Figure 6.** Galleries bored by marine wood borers. (**a**) Tangential cut of a wooden piece showing internal galleries bored by marine molluscs known as 'shipworms' (Teredinidae). Note that galleries follow the fibre direction and are covered by a calcareous lining. Bar = 10 mm. (**b**) Detail of the galleries bored by shipworms showing the calcareous layer deposited by the mollusc in the galleries (arrowhead). Bar = 5 mm. (**c**) Wooden piece attacked by crustaceans known as 'gribbles' (Limnoriidae) and eroded by weathering, leaving internal galleries visible. Note that galleries are superficial and follow any direction. Bar = 10 mm. (**d**) Detail of the exit holes of gribbles in wood. Bar = 10 mm. (**e**) Detail of the tunnels bored by gribbles. Bar = 5 mm. (**f**) External aspect of a wood attacked by shipworms showing entrance holes. Bar = 10 mm.

On the shores of Europe, the most abundant bivalves attacking wood belong to Teredinidae [11]. Shipworms have an elongated vermiform shape and bore galleries of

circular sections in the wood, where they remain during their adult life. They cover the interior of the galleries with a calcareous lining to prevent desiccation (Figure 6a,b,f). The gallery opens to the external environment only at one end, which can be sealed through a pair of calcified plates called pallets that help the mollusc to survive under unfavourable changes in the environment (e.g., during tidal fluctuations) [141]. The animal communicates with the surrounding sea water via a pair of siphons that protrude from the entrance hole. They mechanically bore the galleries in wood (historically, wooden ships were highly damaged, which explains their common name) through the abrasive action of two shells and consume wood as their primary nutrient source with the aid of intracellular bacterial symbionts that colonize their digestive system [141]. However, some species bore in wood and other materials for shelter only. Shipworms are hermaphrodites, and larvae are expelled to the free water, where they are efficiently dispersed by currents before metamorphosis. The adult phase initiates once the larvae adhere to a submerged wood element and start to bore a small gallery. They increase rapidly in size and develop their characteristic wormlike bodies [85]. As they continue to grow, they increase the size of the burrow. Yet, as the entrance holes are only slightly enlarged during their growth, considerable damage may be caused without much external evidence. They can reach a length of several cm to as much as 1 m and a diameter of 3–6 mm up to 25 mm. They usually enter the wood at right angles to the grain and then turn and continue to bore in the grain's direction. Although the use of wood in naval construction has drastically declined in the last century, shipworms still represent an important source of wood deterioration agents in marine construction [137]. Because the abundance and distribution of shipworms can widely fluctuate over time, devastating episodes of wood damage by shipworms can suddenly arise in coastal areas.

Significant damage to coastal wooden constructions can also be caused by small wood-boring crustaceans (around 3–5 mm in length) known as 'gribbles'. In temperate waters, members of the family Limnoriidae are the most prevalent crustaceans attacking wood, although the amphipod Chelura can also cause damage, frequently in co-occurrence with Limnoria [142]. Wood-boring crustaceans damage wood by the tunnels they excavate mainly in the intertidal zone as they can withstand daily exposure to air, although limnoriids have also been found in waterlogged wood at depths over 1000 m [142]. Unlike the tunnels of bivalves, which penetrate deep into the wooden element, the tunnels excavated by gribbles are superficial but, in combination with the erosive action of waves, cause a progressive thinning of the wood structure (Figure 6c,d,e). In the intertidal zone, attacks on wooden piles usually lead to an hour-glass shape. In limnoriids, usually, one pair of individuals live together in a tunnel, the male following the female. The females release their eggs into a ventral pouch, where eggs develop during embryonic development. Juveniles are released into the parental tunnel and frequently excavate side tunnels from the parental one [142]. It seems that limnoriids lack symbiotic microbial flora in their gut [143]. However, the wood they ingest usually contains wood-degrading microorganisms [144], which may facilitate the digestion of cellulose, although other authors suggest that limnoriids produce all the enzymes necessary for lignocellulose digestion [145]. The dispersal of crustaceans occurs during certain periods of the year, but the dispersion is lower than in the case of the larvae of bivalve borers due to the lower number of offspring and their difficulties feeding, while larvae of bivalve borers can feed on plankton. The geographical distribution of limnoriids is largely driven by the water temperature regime, and changes in ocean currents due to global warming may affect the distribution of these species [138]. Unlike wood-boring bivalves, Limnoriids show resistance to wood preservatives, and their control is an unresolved problem. However, recent progress with wood modification (furfurylation) is offering promising levels of protection [146].

### 3. Factors Underlying the Natural Durability of Wood against Biological Deterioration

The natural durability of wood is defined as the inherent resistance of wood to attack by wood-destroying organisms (European standard EN 350:2016). Biological durability is an important factor in the selection of wood or wood products for a particular use. In addition to the climate, product design and use conditions, natural durability is key to the prediction of the service life of wooden products. The natural durability of wood against organisms such as wood-decay fungi and insects can be quantified through standardized tests, whether at the laboratory or field level. These tests are specified in the European standard EN 350:2016 and are specific for the most common deterioration agents. EN 350:2016 classifies the natural durability of wood against fungi into five classes, 1–5, where 1 is very durable and 5 is perishable; against insects into durable and susceptible; and against termites and marine organisms into three classes.

As a biological material, wood is heterogeneous, and its durability varies across species, among populations within species, between individual trees of the same species and even within a given tree. Most of this variability is controlled genetically, while some is due to the environment and cambial age [147–149]. It is difficult to find consistent trends in wood durability among species. Many tropical species are durable, but not all, whereas we can also find very durable species in temperate regions (European standard EN 350:2016). Long-lived species usually have a durable heartwood, the inner part of the wood with no living cells and no reserve material [150], but there are a few exceptions, such as *Picea sitchnensis*, *Fraxinus excelsior* or *Fagus sylvatica*, which may show very active defense mechanisms in the sapwood of living trees to hamper the wound invasion of a decay-susceptible heartwood (European standard EN 350:2016). Species with distinct heartwood are more durable than those with no visual difference between heartwood and sapwood, the outer part containing living cells and reserves in the living tree [150]. The heartwood proportion within species appears to be under genetic control [151], although the environment and management can also be important [149]. For instance, Climent et al. [152] found that the heartwood of *Pinus canariensis* was wider in drier sites, and Margolis et al. [153] reported that severe pruning increased heartwood diameter in *Abies balsamea*. Between functional groups, a meta-analysis [154] showed that gymnosperms generally tend to have lower decay rates than angiosperms, i.e., better durability. These results agreed with the lower decay rates found in temperate gymnosperms compared with temperate angiosperms [155].

The major wood properties involved in wood durability are the amount and composition of heartwood extractives, i.e., secondary low-molecular-weight organic compounds which are produced during heartwood formation, although other factors such as wood anatomy, wood density, lignin content and moisture content are also involved [156,157] (Figure 7). In fact, the durability of different species is determined by different factors. For instance, *Prunus avium* relies mainly on the fungicidal components of the heartwood, whereas the durability of *Entandrophragma cylindricum* depends more on the moisture-regulating components [156]. In wooden products obtained from the same tree, the heartwood is more durable than the sapwood due to the higher concentration of toxic extractive compounds, such as alkaloids, phenols, and terpenes. In general, extractive content increases from the pith towards the outer heartwood and reaches a maximum at the transition zone between heartwood and sapwood; thus, as the tree ages, the extractive content increases [158], and the decay susceptibility decreases [159]. Longitudinally, extractive concentration decreases with tree height. In many species, the outer heartwood at the base of the tree is the most resistant to decay [157,160]. Durability gradients may reflect biological detoxification, natural oxidation of extractives, continued polymerization of extractives to produce less toxic compounds or increases in extractive deposition with age [158,160].

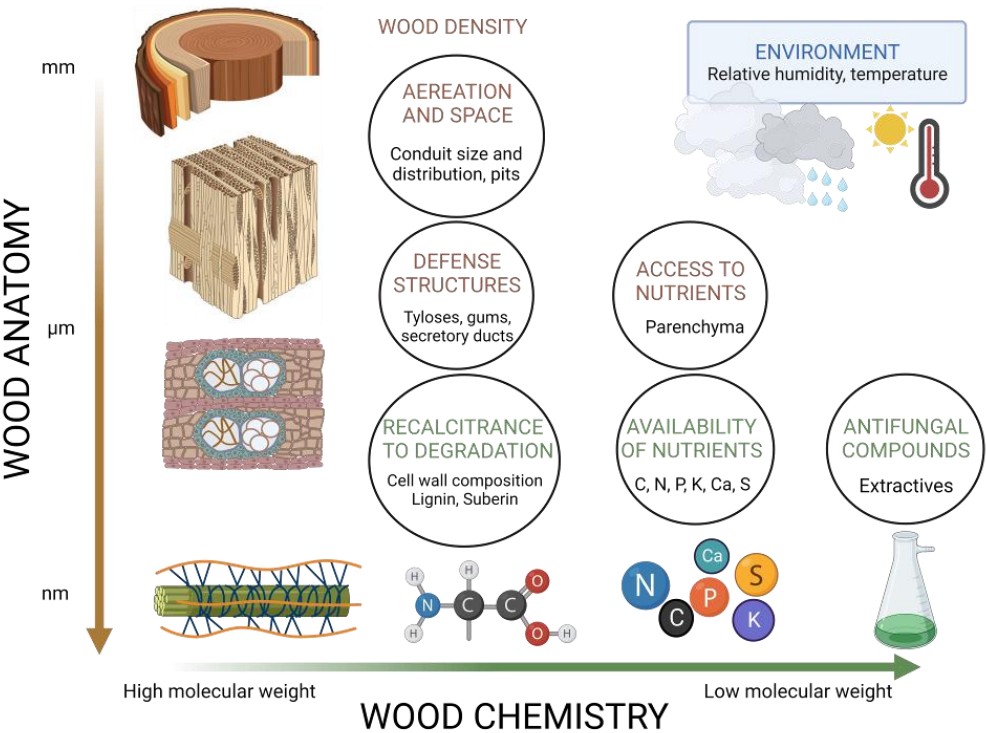

**Figure 7.** Anatomical, chemical and environmental factors involved in natural durability of wood. According to the vertical axis, factors vary from macroscopic to microscopic and molecular scales. Created with BioRender.com.

### 3.1. Wood Anatomy and Wood Density

Microscopic wood anatomy (Figure 8) and cell wall ultrastructure have a central role in regulating biodeterioration [144,161]. Differences in wood anatomy between gymnosperms, where up to 95% of the xylem consists mainly of tracheids, with the rest being parenchyma cells and resin ducts, and angiosperms, with more specialized cell types, larger conducting elements and a higher proportion of parenchyma cells, are in part responsible for differences in their natural durability [18]. The lumen of the conducting elements, the structure, morphology and composition of pits, the parenchyma fraction and the presence of secretory structures and tyloses seem to determine, to some extent, the rate of decay [162]. In a meta-analysis of 83 sites from all forested continents with 90 angiosperms and 108 gymnosperms comparing the durability of different species decomposing in the same environmental conditions, the decomposition rate was more than 130% higher in angiosperms than in gymnosperms [154]. In addition to physical and chemical barriers to the spread of decay, the moisture content of wood is of vital importance for wood-decay agents and also depends on some anatomical traits. The irreversible closure of pits, conduit blockage by tyloses and gums, and the size and distribution of vessels in angiosperms affect the rate of moisture uptake and release, favoring or hampering decay spread. For instance, wood with large and frequent vessels can absorb more water than denser wood with fewer vessels [163]. The size of the pit aperture affects the desorption rate, and species with small pit openings or vesture pits hold water longer than those with larger pits [163] (Figure 8g,h).

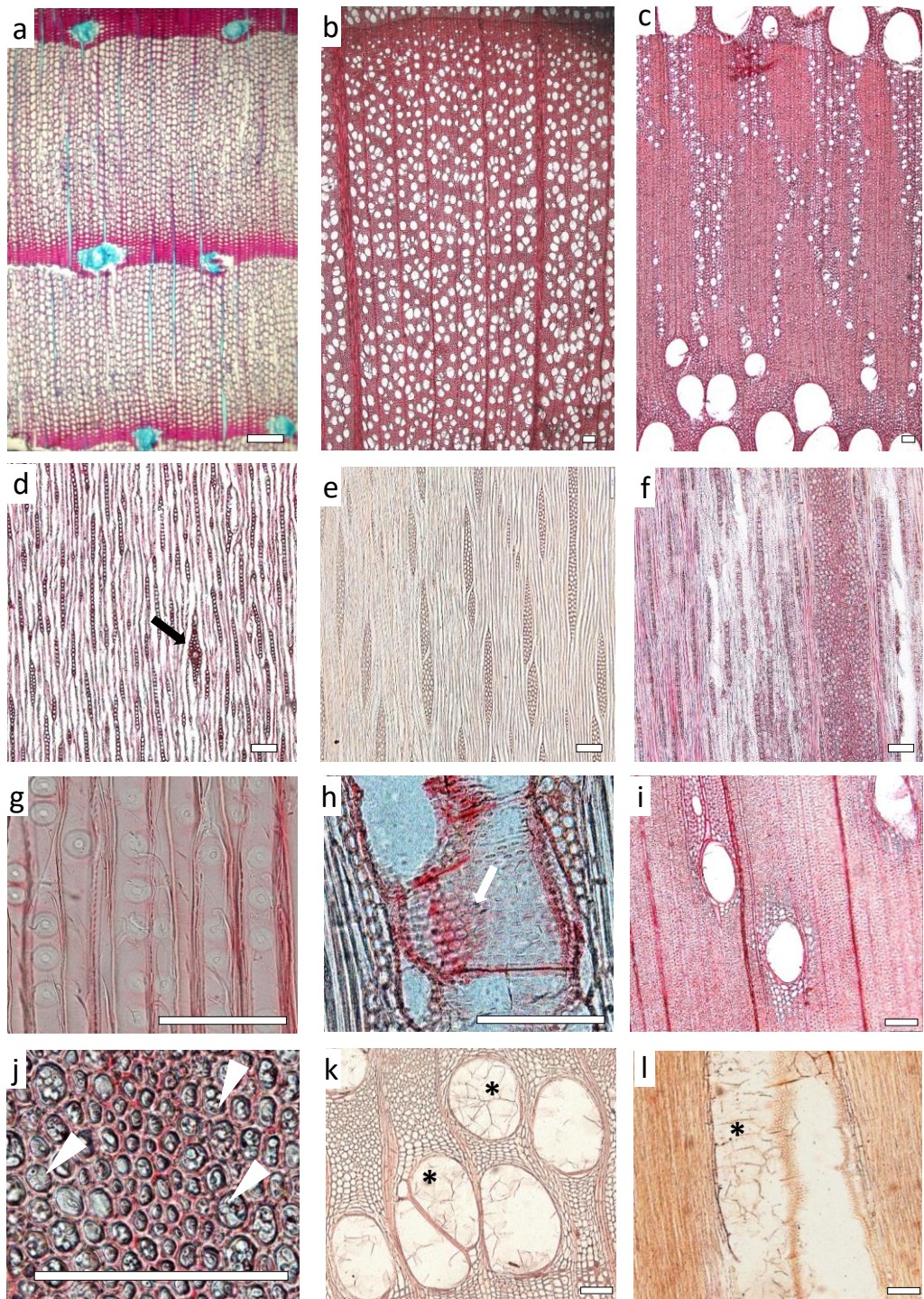

**Figure 8.** Wood anatomical features related to species' natural durability. (**a–f**) Differences in conduit diameter and density, proportion of parenchyma and presence of secretory canals between gymnosperms and angiosperms. (**a**) Cross-section of *Pinus pinaster*; xylem is formed mainly by tracheids, with parenchyma in uniseriate rays and axial resin canals in blue. (**b**) Cross-section of *Fagus sylvatica*, a diffuse-porous species, with vessels grouped radially and multiseriate rays. (**c**) Cross section of *Quercus pyrenaica*, a ring-porous species with uniseriate and wide multiseriate rays and abundant axial parenchyma. (**d**) Tangential section of *Pinus pinaster*; black arrow points to a radial resin canal. (**e**) Tangential section of *Robinia pseudoacacia* with short multiseriate and uniseriate rays. (**f**) Tangential section of *Quercus pyrenaica*; note the large multiseriate rays and short uniseriate rays. (**g**) Areolate pits between tracheids in *Pinus pinaster*. Aspirated pits hamper fungal spread. (**h**) Pits between vessels in *Quercus pyrenaica*. Species with small pit openings hold water longer than those with larger pits. (**i**) Paratracheal parenchyma in *Eucalyptus globulus*. Higher

proportion of parenchyma increases decay susceptibility. (**j**) Abundant starch (arrow heads) in parenchyma rays. (**k**) Blockage of vessels with tyloses hinders wood accessibility for microorganisms. Asterisks show tyloses in *Robinia pseudoacacia*. (**l**) Detail of tyloses (asterisk) in a tangential section of *Robinia pseudoacacia*. All bars = 100 μm.

### 3.1.1. Vessels and Fibers

The distribution and ratio of wood cell types and the presence of intercellular spaces affect the penetration and colonization ability of fungal hyphae and determine whether the fungus can tap into the nutrient sources and have enough oxygen within the wood. In gymnosperms (Figure 8a), hyphal growth and the diffusion of degradative substances are faster within the larger cell lumina of earlywood tracheids and parenchyma than in latewood tracheids [164]. In angiosperms (Figure 8b,c), the proportion of vessels, fibers and parenchyma may likely influence decay rates. The high durability of *Pterocarpus soyauxii* and species from a semi-deciduous forest in Ghana was linked to a low vessel:fiber ratio [156,165]. Fibers have small lumens and thick cell walls and usually comprise most of the deterioration material. However, due to the relatively low diffusion of gases in their lumen, the metabolic activity of decay for some agents may be restrained [166]. For instance, fibers are rapidly degraded by brown-rot fungi but are resistant to decomposition by white-rot fungi [18]. The arrangement of fibers determines the tendency to splinter or shatter, increasing the surface exposed to decomposer organisms and hastening the decay process [167]. Only tension-wood fibers are associated with soft-rot cavities in *Fagus sylvatica* attacked by *Meripilus giganteus* [168]. On the contrary, large vessels may provide favorable microsites for fungal activity due to their higher moisture and oxygen contents and accessibility for fungal hyphae [169]. Although decay fungi are able to move through pit membranes between conducting elements, which are subsequently enlarged, aspirated pits in gymnosperms, i.e., when the torus of the pit moves across the pit chamber to the pit border, blocking the connection between tracheary elements, or pit encrustation with extractives, reduces water and hyphal movement and presumably creates conditions unfavorable to decay [149].

Vessels are often plugged with tyloses and gums (Figure 8k,l), particularly in the heartwood, which hinders accessibility for microorganisms. The transition from sapwood to heartwood is marked by the death of parenchyma cells and vessel occlusion with tyloses or gums and the aspiration of pits. These physical barriers may prevent or slow down the spread of pathogens and wood-decay agents, as well as compartmentalize the xylem after wounding in the living tree [170]. Tyloses are overgrowths of the protoplast of adjacent living ray parenchyma or paratracheal parenchyma cells (Figure 8i) into vessels and occasionally tracheids and fibers. They plug vessels, which can no longer conduct water, and can contain a wide variety of organic and mineral compounds, including gums, resins, starch, crystals and suberin [170]. Some fungi cannot penetrate occluded vessels and expand tunnelling through cell walls, while others, by contrast, preferentially degrade the polyphenolic occlusions in the cell lumina [164]. Gums include several secondary metabolites with variable chemistry and can be of chemotaxonomic interest [158]. Gums have been reported to form after the production of gels where pathogens have released pectinolytic and cellulolytic enzymes [171]. The high content of gums filling cell lumens in the heartwood of *Prosopis africana* limited the penetration of water and was related to its high natural durability [172].

### 3.1.2. Parenchyma

The amount of parenchyma in the xylem also affects natural durability. In particular, wood ray height and ray thickness (Figure 8d,e,f) were positively related to the wood decay rate in temperate species [155]. Parenchyma rays store and radially transport water, small molecules and nutrients so that more parenchyma implies a higher concentration of these compounds, including nitrogen, within the tree, which in turn could hasten the decay rate [173]. Parenchyma also facilitates the colonization of wood by hyphae and

improves their radial distribution within the xylem [18]. Fungi and bacteria usually start wood colonization from the rays, and later, they enter other parts of the wood via pits or from cell to cell, boring holes through cell walls [18]. Parenchyma cells are not lignified and contain significant amounts of easily assimilable nutrients (Figure 8j); thus, fungi and bacteria with the capacity to degrade cellulose or hemicelluloses can attack these cells. In fact, some fungi, such as molds and blue-stain, only attack these non-lignified cells but normally do not cause major wood decay [162]. On the other hand, living parenchyma cells are essential in tylosis and gum deposition during heartwood formation or as a defense mechanism in the sapwood, occluding the vessels, slowing down and preventing the spread of pathogens and eventually decreasing decay rates. The success of vessel-blocking as a barrier against pathogens depends on the presence of suberized walls and the accumulation of anti-microbial compounds such as tannins, catechol, flavonoids and coumarins [174]. The amount of parenchyma is associated with higher resistance to brown-rot fungi, apparently due to the cell wall's morphology, but with lower resistance to white-rot fungi, which preferentially degrade parenchyma cells during the early stages of decay [18].

### 3.1.3. Resin Ducts and Other Canals

Some species show xylem ducts filled with phytochemical compounds, which serve as a secondary defense against colonization by fungi [175]. Perhaps the most studied structures are the resin canals of conifer species [176]. Resin canals are abundant secretory structures in the axial and radial direction consisting of ducts surrounded by epithelial cells, which fill the duct with a complex mixture of terpenoids, phenolic compounds and fatty acids, with several functions, such as pathogen and pest repellents or biocides [177]. In some angiosperms, secretory cavities and secretory ducts also contain volatile compounds which exert toxic effects and decrease the rate of decay [178]. Genera of the families Anacardiaceae and Burseraceae show radial canals, whereas some Caesalpiniacieae or Dipterocapaceae show axial canals [150]. Chemicals contained in canals in angiosperms are more diverse than in gymnosperms and include gums, phenols, proanthocyanidins, latex, etc. [179], as mentioned below.

### 3.1.4. Wood Density

Wood density, which is related to the number of voids and cell walls in the xylem, is commonly considered a major trait over wood decomposition rates [180,181]. However, experimental studies and meta-analyses show disparate results. In a long-term study with 15 tropical species in Bolivia, wood density was not related to decomposition rates [182], though in the Amazon, several works [183–185] found a direct correlation between wood density and decomposition. In temperate angiosperms, decay rates were higher in denser wood, while gymnosperms showed the opposite pattern [155]. Wood density seems to be more determinant when wood is attacked by termites. These insects tend to feed on lower-density species with low concentrations of extractives when possible [186,187].

### *3.2. Wood Chemistry*
### 3.2.1. Cell Wall Composition

Wood cell walls are the main substrate for the growth of decay fungi [179]. Wood cell walls are organized in layers of different thicknesses and compositions. Chemically, all cell walls are composed of cellulose, hemicelluloses, lignin and sometimes also pectins [188]. Although the quantity of cellulose remains between 40% and 50% in both angiosperms and gymnosperms, the lignin content is higher in gymnosperms (28%–32% vs. 16%–26% in angiosperms), whereas the hemicelluloses are higher in angiosperms and differ in composition [189]. Angiosperms have more glucuronoxylan and galactoglucomannan in their hemicellulose, which could be related to hindering brown-rot fungi ac-

tivity in angiosperms [169]. The availability of cellulose and hemicellulose seems to be essential for decay within wood cell walls since most fungi and bacteria show cellulase and/or hemicellulase activity [190]. When lignin impregnates the cell wall, it forms a more impenetrable matrix and makes cellulose less accessible and more resistant to biodegradation [157]. Lignins are a heterogeneous group of compounds, and their composition influences wood decomposition by fungi [18]. In gymnosperms, the lignin monomer is guaiacyl, whereas angiosperms have both guaiacyl and syringyl [191]. The high lignin content and proportion of guaiacyl give wood higher protection against decay organisms, mainly white-rot fungi [161]. Guaiacyl is difficult to oxidize, and it is very compact, which prevents enzymatic attack and partly explains the better durability of gymnosperms compared to angiosperms and the lower decay rates in the latewood of gymnosperms than in earlywood [18,161,192]. As an example, *Picea abies* and *Pinus sylvestris* were not durable against *Coniophora puteana*, a brown-rot fungus, but they showed high durability against *Trametes versicolor*, a white-rot fungus [156,193].

In angiosperms, the distribution of syringyl and guaiacyl in different cell types may have a major influence on the speed and pattern of decay by fungi and bacteria. Syringyl is most common in fibers and parenchyma, whereas vessels have higher concentrations of guaiacyl [18]. In ring-porous species, such as *Quercus robur*, fiber tracheids also have higher guaiacyl content than fibers [194]. Consequently, vessels and fiber tracheids have been found to be more resistant to soft-rot and white-rot fungi [195], while wood with abundant libriform fibers was highly susceptible to soft-rot [164].

In spite of the low concentration of pectin in wood, the hydrolysis of this compound from the tori of bordered pits is considered a key step in the colonization of wood by brown rot fungi [196]. Some white-rot fungi also degrade the pectin in the cell wall. *Meriplius giganteus* preferentially degrades this compound in the parenchyma rays of *Fagus sylvatica* [197]. Pectin hydrolysis by brown and white fungi is related to the production of oxalic acid, which solubilizes pectins by chelating calcium [198].

### 3.2.2. Nutrients

The low availability of mineral nutrients in virtually all wood types constrains decomposition and increases natural durability. In general, gymnosperms have lower concentrations of N, P, K and Ca than angiosperms [173] and decompose more slowly. Of all nutrients, N seems to be the most determinant in wood decomposition [154]. The C/N ratio of *Betula* sp. is 55, and in *Acer pseudoplatanus*, it is as high as 401 [18]. Consequently, wood with higher concentrations of N and P and a lower lignin:N ratio decomposes faster than wood with a high C/N ratio [154]. Cell wall decay is related to the production of lignocellulase enzymes, rich in N, produced by fungi and bacteria. Although decaying fungi can consume large quantities of carbohydrates and lignin using relatively small amounts of N, when the concentration of nutrients in the wood is low, microbes divert energy from lignocellulase synthesis towards the acquisition of N and P, and thus, the rate of decomposition becomes slower [154]. S and K have also been positively related to the decomposition rates of deadwood logs of temperate species [155]. Besides N and P, S seems to be one of the limiting macronutrients for fungal growth, probably because it is essential for two amino acids and various cofactors [155]. Moreover, during the decay process, changes in the nutrient status of the wood must also be considered since they may influence the fungal flora [199].

### 3.2.3. Extractives

Several species show high wood durability due to the presence of extractives, particularly in their heartwood [157,158]. The importance of extractives is such that in some species with durable wood, if extractives are removed, they become susceptible to decay [157,200]. Alternatively, the impregnation of extractives from durable species to decay-prone species or non-durable sapwood improves their durability [200]. Termites generally avoid wood with high concentrations of extractives [201,202]. The distribution

of extractives also influences durability. Extractives that are impregnated into the cell wall are more effective in preventing microorganism growth than are those in the cell lumen [158,203]. Some phenolic compounds produced during heartwood formation, for example, polymerize with preexisting cell wall components and decrease accessibility by pathogens [204].

Extractives include a large variety of non-structural compounds with diverse chemical compositions, commonly of hydrophobic nature, which can be extracted using several solvents. Some extractives have antimicrobial and insecticidal properties, whereas others influence moisture dynamics or influence other wood properties such as density, color, odor, or flammability [179,185,205,206]. Extractives are normally classified into four groups: phenolic compounds, terpenoids, alkaloids and fatty acids [207]. Their fungicidal mechanisms are diverse; some interact with fungal enzymes, while others show antioxidant activity or disrupt cell walls and membranes, producing leaks in the cell content of microbes and affecting ion homeostasis [208–210]. Since the mode of action and molecular size of enzymes of white-rot and brown-rot fungi are different, many fungicidal components are only effective against a few fungal species [211].

The chemical nature of wood extractives may be more important than the amount of extractives, per se, except for certain highly fungicidal extractives such as thujaplicin, the most toxic of the tropolones, found in Cupressaceae [211], which are strong metal chelators that act as uncouplers of oxidative phosphorylation and prevent the decay of lignocellulose [206,212]. *Larix* spp. produce large amounts of extractives, but with little protection to decay [213]. In other cases, multiple low-toxicity extractives can act synergistically, as in *Tectona grandis*, where durability is the result of the action of various extractives, predominantly quinones, such as anthraquinones and tectoquinones [214,215]. In a study with three durable subtropical species, *Bocoa prouacencsis* showed the highest durability with the lowest extractive content due to the high fungicidal activity of its extractives [185]. Additionally, Van Geffen et al. [182] concluded that the inhibiting factors of phenolic extractives on wood decomposition were related to their chemical nature and not to the quantity present in the wood, possibly due to the detoxifying activities of some enzymes, such as laccases and tyrosinases, secreted by some decomposing fungi.

Many gymnosperms show specialized resin secretory structures. The genera *Pinus* and *Picea* develop a densely interconnected network of axial and radial resin canals capable of transporting resin several meters [216]. The capacity of gymnosperms to produce resin has been related to deterring insect pests and resistance to fungi [217], as in *Heterobasidion annosum* [218], *Fomes annosus* [219] or *Ceratocystis polonica* [220]. Resin consists mainly of terpenoids, both volatile terpenes and non-volatile resin acids, phenolic compounds such as lignans and tropolones, and fatty acids, which slow down fungal growth and decay activity [169,179]. Terpenoids are toxins and feeding deterrents to many pests and pathogens [217]. The monoterpene limonene extracted from *Citrus limon* was involved in the increase in durability of *Pinus sylvestris* against wood-decay fungi [221]. A type of triterpene, saponins, has detergent properties and may disrupt cell membranes after being absorbed [222]. In angiosperms, the chemical composition of extractives, including resin-like substances, is more diverse. Latex from *Hevea brasiliensis* and *Palaquium gutta* includes proteins, alkaloids, oils, tannins, polysaccharides and phospholipids. In *Prunus* and *Acacia*, carbohydrate gums are common, whereas mansonones are common in the heartwood of *Ulmus* [179]. In Eucalyptus, the concentration and composition of extractives defined two groups, one with red heartwood, including *E. marginata*, *E. camaldulensis* or *E. grandis*, and another with brown heartwood, with *E. globulus* or *E. dumosa* [148].

Phenolic compounds are a heterogeneous group that includes several complex molecules involved in wood durability, such as lignin, tannins, stilbenes, quinones or flavonoids. Some scavenge radicals and active oxygen, including singlet oxygen, free radicals, and hydroxyl radicals, which reduce the effectiveness of fungal oxidative enzymes [223]. Tannins act as feeding repellents, which can bind proteins, inactivate her-

bivore digestive enzymes and create complex aggregates which are difficult to digest. Moreover, they are abundant in the heartwood of some species (for instance, gallic acid, catechins, and salicins), preventing fungal and bacterial decay [179]. Quinones are highly reactive molecules, and stilbenes, such as pinosylvin in *Pinus densiflora*, inhibit the growth of *Heterobasidion annosum* [224].

Most alkaloids are now believed to function as defenses. As some phenolic compounds do, alkaloids scavenge radicals and delay the initial stages of fungal decay [225,226]. In *Dicorynia guianensis*, from French Guyana, alkaloids exhibit dose-dependent fungistatic activity against *Pycnoporus sanguineus* and *Trametes versicolor* [227]. The growth of white-rot *Trametes versicolor* and brown-rot *Oligopous placentus* was inhibited by extractives of *Lantana camara*, with abundant alkaloids [228], and the high durability of *Erythrophleum fordii* has been partly attributed to these chemicals [229].

Suberin is resistant to microbial decay, and it is highly hydrophobic. It appears in the heartwood of some species, such as *Quercus robur* [230], in the resin canals of some *Pinus* [231,232] and in tyloses [170]. Fatty acids, besides their antibacterial, antifungal and antioxidant properties, as in resins of Pinaceae (oleic, linoleic, pinoleic, etc.), are deposited in the cell lumen and pit membranes, where they decrease permeability [233] and deter fungal growth [234]. However, some fatty acids and sterols are responsible for reductions in wood durability since they are nutrients for fungi [235].

### 3.3. Environmental Effects

Environmental conditions, specifically humidity and temperature, exert strong control over wood durability and decomposition rates [236]. The wood decay rates of living trees in tropical and sub-tropical sites are faster than in temperate zones [237]. An increase in ambient temperature and increase in winter precipitation promoted wood decay in North American forests [238]. In a global study of decomposition rates, Meentemeyer [239] found that actual evapotranspiration from subpolar to temperate forests was a key factor affecting decomposition rate.

### 3.3.1. Moisture Content

The relative humidity and temperature of the atmosphere influence wood moisture content and consequently trigger wood swelling or expansion and, on the other hand, affect susceptibility to the establishment of wood-degrading organisms [199,240]. Mold fungi usually occur when ambient relative humidity ranges between 80% and 95% [241]. Most rot fungi cannot grow if the wood moisture content is below the fiber saturation point, ca. 30%, since below this point, free water for extracellular transport of fungal metabolites is no longer available, and only water tightly bound in the cell walls remain, although certain fungi can survive for many years in the dry wood [242]. Conversely, fungal growth is hampered when moisture content exceeds 80% because of reduced oxygen availability [79]. Therefore, dry wood or water-saturated wood seldom decays. In this sense, the storage of logs under sprinkling water has been traditionally used as a wood protection method in forestry [243], and archaeological studies have recovered wood buried in ocean sediments for centuries [244]. In these situations of high humidity and reduced oxygen, such as waterlogged wood, boats, or logs under constant water sprays, wood is commonly attacked and degraded by soft-rot fungi and tunneling bacteria, whereas white and brown rot are, in general, more sensitive to reduced air supply [79,245]. For other degrading agents, such as termites and beetles, some moisture is also needed. Particularly, subterranean termites (Rhinotermitidae) require humid environments and higher wood moisture content than dry-wood termites (Kalotermitidae). *Reticulitermes flavipes* cannot survive on wood with less than 16% moisture [246]. The highest foraging activity in *Coptotermes formosanus* has been found for moisture contents of 25%–50% and up to 103%. However, if the moisture content is higher than 133%, both *Coptotermes formosanus* and *Reticulitermes flavipes* decrease feeding activity [247]. In fact, termites can manipulate wood moisture levels by wrapping pieces of wood in a layer of

clay, thus increasing moisture content and facilitating foraging [248]. If moisture levels in the air are also high, flying reproductive individuals can initiate new colonies [240]. Dry-wood termites (Kalotermitidae) nest in wood with low moisture content [6]. The larvae of wood-boring beetles require wood moisture contents between 13% and 30% and a relative humidity of 65%–90% for viable infestation [249]. The larvae of *Hylotrupes bajulus* developed optimally in wood with moisture content between 15%–25% [250]. Susceptibility to beetle infestation decreases with ventilation, dehumidification and insulation since they lower wood moisture content [240]. Insects that do not feed on wood but rather use wood as a habitat, such as carpenter ants, *Camponotus* spp., also prefer wood with high moisture content [240]. Thus, the moisture-absorbing and -desorbing properties between woods of different species or between sapwood and heartwood may significantly impact wood moisture content under changing environmental conditions and, consequently, affect susceptibility to biodeterioration. Some species, such as *Thuja plicata* and *Callitropsis nootkatensis*, prevent or delay fungal and insect growth by taking up liquid water more slowly and desorbing more rapidly than others, such as *Pseutotsuga menziesii* or *Pinus* sp. [251].

Some studies report that local climate conditions have a major influence on the biological decomposition of wood [252,253], while others state that the influence and nature of the soil and the microorganisms present in the matter pose a higher threat than climate conditions do [254,255]. In any case, soil contact usually facilitates fungal colonization due to the easy accessibility and higher moisture content of wood [169], and heavy rainfalls or flooding combined with elevated temperatures increase termite activity [248]. Accordingly, significantly higher decomposition rates are shown for plant material in contact with soil, which is use class 4 for solid timber and wood-based products (European standard EN 335:2013), as in fence posts, poles and timbers of cooling towers, compared to those that are not in soil contact [256,257].

### 3.3.2. Temperature

Besides adequate wood moisture, oxygen and nutrient availability, most decay fungi require temperatures above freezing to survive and proliferate, although spores can withstand long periods of freezing and periods of repeated freezing and thawing [242]. Fungal metabolism is strongly influenced by temperature. Metabolic reaction rates mediated by enzymes increase with increasing temperature until a certain point when some reactions become thermodynamically limited or heat affects protein stability. In fact, global studies have shown that wood decay increases twice every 10 °C increment, driven in part by enzyme kinetics [258]. Fungi can withstand temperatures between −20 °C and 40 °C, and the lethal temperature is around 65 °C. The optimum temperature for rot fungi activity ranged between 22 °C and 36 °C, although there are some exceptions. For instance, *Serpula lacrymans* cannot survive temperatures above 37 °C, and its optimum activity is around 18 °C–20 °C [259]. Termite wood feeding is very sensitive to temperature, with consumption increasing 6.8 times every 10 °C [260]. Their optimum feeding temperature ranges between 26 °C and 32 °C, with a relative humidity of the air of 70%–90%. Therefore, they are mostly restricted to tropical, sub-tropical, and temperate climates [6], although with climate warming, termite wood decay will likely increase as termites access more of Earth's surface [260].

Finally, other factors, such as fertilization or management, can influence wood durability since they affect the environment of tree growth, which can alter the extractive content and nutrient concentration [261]. For example, fertilizers can increase levels of nitrogen and phosphorus in wood and promote fungal attack as described above but may also increase the amount of reserves for the synthesis of fungitoxic compounds by the plant [262].

## 4. Conclusions

Although wood is a recalcitrant material inaccessible as a food resource for most living organisms, a number of xylophagous agents have developed highly specialized enzymes to debilitate the complex structure of the cell wall and digest wood polymers. Most wood-deteriorating agents produce endogenous wood-degrading enzymes, but in certain cases, symbiotic associations among different organisms are needed to successfully metabolize wood components or to provide nutrients that are naturally scarce in wood. Xylophagous fungi are the organisms causing the greatest economic impact on wooden structures worldwide, but certain species of termites, saproxylic beetles and marine borers can also cause serious losses at local or regional scales. Global trade and climate change are inducing a shift in the distribution of invasive organisms with the potential to cause damage to wooden elements, a trend that will probably be exacerbated in the next decades. Policies preventing the accidental introduction of invasive organisms should therefore be reinforced. There are still important knowledge gaps regarding the mechanisms that wood-deteriorating organisms use to attack wood, their ecology and mode of dispersion, and of some wood traits affecting the natural durability of wood in service. To improve the social perception of wood as a raw material, further research is also needed to develop or improve environmentally friendly methods for preserving wood species of low natural durability against biological deterioration.

**Author Contributions:** Both authors (J.A.M. and R.L.) contributed equally to this work. All authors have read and agreed to the published version of the manuscript.

**Funding:** This research received no external funding.

**Institutional Review Board Statement:** Not applicable.

**Informed Consent Statement:** Not applicable.

**Data Availability Statement:** Not applicable.

**Acknowledgments:** We thank the academic and technical personnel of ETSI Montes, Forestal y del Medio Natural (UPM), who contributed to the collection and maintenance of the wood samples shown in the figures.

**Conflicts of Interest:** The authors declare no conflict of interest.

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
