# Peer review of "Biological Deterioration and Natural Durability of Wood in Europe"

_forests, doi:10.3390/f14020283_

Round 1

Reviewer 1 Report

Dear Authors,

Please see the attached word file.

Kind regards

Reviewer 2 Report

The topic of the research work and manuscript is really interesting and provides new information. However there are several issues to be addressed towards its quality improvement before publication. 

Instead of "woods", you rather use "wood species" in the whole text. In line 13, instead of machinery, you rather use "mechanism".  In line 19, instead of "enough", you rather use "adequate". In line 20, instead of "attack", you rather use"catastrophic action". General a thorough check on syntactical and grammatical errors or less targeted choice of terms in the text, seems to be imperative. You did not follow and apply the appropriate format in the whole text. You rather combine the paragraphs of the different references. In lines 61-64, provide a reference. In line 68, how are the soil properties. In line 73, you rather provide the chemical composition of wood, the monomers, oligomers, polymers (cellulose lignin etc.) and the role of them. The use of several terms seems ackward (for example: array of microorganisms?). The term "hemicellulose" should be used in plural. In lines 80-90, you did not refer to those microorganisms that prefer both cellulose and lignin, or prefer lignin even more (there are such species). Generally, the knowledge you provide are much generic, already very well known and accepted and I am wondering where is the new knoledge and the contribution of this manuscript (compared to one of the many relevant scientific books). Table 1 seems interesting and useful though. In the caption of figure 1, you could refer only to the common name of the wood species or if you keep the scientific name you should turn them in italics, as well as the species of fungi. In line 236, what is that Bar=5mm? (it is not very clear). In line 244, I believe that the "Ascomycota" needs to be corrected to "Ascomycetes". In 305 line, corect the "in the in the". In 310, the "Bacterial richness" probably changed to "variety". In 832-833 lines, incorporate as well the relevant work of https://doi.org/10.3390/f10121111 . In line 969, the word "growth" needs correction. In 1045, the "machinery" as well.

Reviewer 3 Report

This is a really well-written paper and I enjoyed reading it.

Some comments for improvements:

Line 54 - than or that?

Table 1 - Please keep only the first sentence as title, the remaining should put as the footnote of the table.

While the content is sufficient and very informative, I find that the there are lacking of table and figure in the paper, which is very important as it could attract the attention of reader more easily and digest the information more easily. I suggest the author to insert more Table and Figure in the paper:

1. An figure conclude overall subtopic 2.0 and 3.0

2. Some tables that list out the species of biodeteriorating agent and species affected is beneficial.

Round 2

Reviewer 1 Report

Thank you for the extensive changes. I accept the manuscript in its revised form. 

Reviewer 2 Report

As I have checked the authors have implemented the proposed changes in the revised verion of manuscript towards the improvement of their work. Almost all the changes have been implemented and in my opinion, the manuscript is well-prepared and organized enough to be accepted for publication in this journal.